# Artificial Neural Networks in the Prediction of Genetic Merit to Flowering Traits in Bean Cultivars

**Renato Domiciano Silva Rosado [1]**, **Cosme Damião Cruz [1,2]**, **Leiri Daiane Barili [3]**,
**José Eustáquio de Souza Carneiro [2]**, **Pedro Crescêncio Souza Carneiro [2]**,
**Vinicius Quintão Carneiro [4]**, **Jackson Tavela da Silva [2]** and **Moyses Nascimento [1,2,\*]**

[1] Department of Statistics, Graduate Program in Applied Statistics and Biometry,
Federal University of Viçosa (UFV), Viçosa 36570-900, Brazil;
rosado.rds@gmail.com (R.D.S.R.); cdcruz@ufv.br (C.D.C.)

[2] Department of General Biology, Graduate Program in Genetics and Breeding, UFV, Viçosa 36570-900, Brazil;
eustaquiocarneiro@yahoo.com.br (J.E.d.S.C.); pcscarneiro@gmail.com (P.C.S.C.);
jacksontavela@gmail.com (J.T.d.S.)

[3] Faculdade Centro Mato Grossense (FACEM), Sorriso 78890-000, Brazil; leyridaiana@hotmail.com

[4] Department of Biology, Graduate Program in Genetics and Plant Breeding,
Federal University of Lavras (UFLA), Lavras 37200-900, Brazil; vqcarneiro@gmail.com

\* Correspondence: moysesnascim@ufv.br

**Abstract:** Flowering is an important agronomic trait that presents non-additive gene action. Genome-enabled prediction allow incorporating molecular information into the prediction of individual genetic merit. Artificial neural networks (ANN) recognize patterns of data and represent an alternative as a universal approximation of complex functions. In a Genomic Selection (GS) context, the ANN allows automatically to capture complicated factors such as epistasis and dominance. The objectives of this study were to predict the individual genetic merits of the traits associated with the flowering time in the common bean using the ANN approach, and to compare the predictive abilities obtained for ANN and Ridge Regression Best Linear Unbiased Predictor (RR-BLUP). We used a set of 80 bean cultivars and genotyping was performed with a set of 384 SNPs. The higher accuracy of the selective process of phenotypic values based on ANN output values resulted in a greater efficacy of the genomic estimated breeding value (GEBV). Through the root mean square error computational intelligence approaches via ANN, GEBV were shown to have greater efficacy than GS via RR-BLUP.

**Keywords:** common beans; multilayer perceptron; radial basis function network; genomic prediction

## 1. Introduction

The development of common bean cultivars contributed significantly to the increase in the mean national yield of 500 kg ha$^{-1}$ (in the 1970s) [1] for more than 1331 kg ha$^{-1}$ (in the 2018/2019 season) in the mean of three planting seasons (first crop or water (1504.50 kg ha$^{-1}$), second crop or drought (1492 kg ha$^{-1}$) and third crop or irrigated (996.5 kg ha$^{-1}$), seeing current Black, Carioca and other grain color patterns of common beans cultivars [2]. The progress in grain yield, productivity components, grain technological quality and nutritional quality, is mostly attributed to genetic improvement [3,4]. Besides these, flowering time traits, for example, days to flowing (DTF) and days to first flower (DFF) presents importance in a breeding program of common bean. The identification of cultivars with an early cycle allows the planning of harvests for periods of less rain, the reduction of water consumption by irrigated crops, and reduction of the time exposed to the risk of plague and disease [5–7].

Meuwissen et al. [8] introduced Genome Selection (GS) aiming to aggregate information on molecular markers and phenotypes in the prediction of individual genetic merit. GS has been successfully used to

accelerate genetic progress in plant breeding [9]. However, the statistical modeling in the GS approach generally faces some difficulty due to the high dimensionality and multicollinearity. Another challenge faced by GS refers to modeling the intra-and inter-allelic interactions. These non-additive effects, if not considered in the model, can reduce the predictive ability of these models affecting ranking breeding values [10].

Aiming to consider the non-additive effects in the model fitting, Gianola and van Kaan [11] presented a theoretical perspective of Reproducing Kernel Hilbert Spaces Regression (RKHS) methods for genomic prediction. Toro and Varona [10] quantified the efficiency of addingdominance in the GS fitting. de Almeida Filho et al. [12] proposed different approaches to considerer the non-additive effects using semi and non-parametric methods.

Another approach that can be used to capture and model the non-addictive effects, increasing the predictive performance of the model, is the use of an artificial neural network (ANN). The ANNs recognize patterns and regularities of data and represent an alternative as a universal approximation of complex functions [13]. In a GS context, this feature allows automatically to fit factors such as epistasis and dominance since it is not necessary to know a priori if the data have these effects [14]. In addition, this approach does not require any assumptions about the distribution of phenotypic values as the statistical methods do. ANNs have been used successfully in several breeding studies to predict the genetic merit using simulated [15,16] and real data [17–19]. Overall, these studies show that the application of ANN in GS presents great potential for capturing complex interactions since the accuracy values and the bias are, respectively, higher and lower compared with those obtained through traditional GS methodologies (for example, G-BLUP).

According to Krause et al. [20] and Nayak et al. [21], flowering traits in bean cultivars present non-additive gene action. Therefore, to use an approach, such as ANN, that allows modeling the non-additive effects automatically seems interesting to predict the genetic merit of those traits.

The objectives of this study were to predict the individual genetic merits of the traits associated with the flowering time in the common bean using ANN approaches, and to compare the predictive abilities obtained for ANN and Ridge Regression Best Linear Unbiased Predictor (RR-BLUP) [8] for predicting genetic merit.

## 2. Materials and Methods

### 2.1. Experiment and Experimental Material

The phenotypic and genotypic data were provided by Beans Breeding Program at Plant Science Department of Federal University of Viçosa, Minas Gerais, Brazil. The experiments involving 80 bean cultivars, divided into two groups, Carioca and Black, recommended between 1960 and 2013 by research institutions in Brazil (Embrapa, IAC, UFV, IAPAR, Epamig, UFLA, Fepagro, Epagri, FT Seeds), which were selected through scientific records (articles in indexed journals) as well as experience reports from breeders of different breeding programs [22].

Four experiments were established. One located in Viçosa/MG (lat 20°45′14″ S, long 42°52′55″ W, alt 648 m asl), and the other in Coimbra/MG (lat 20°51′24″ S, long 42°48′10″ W, alt 720 m asl). Cultivars were planted at each location in the dry-summer (February) and winter (July) seasons of 2013, following a randomized complete block design with three replicates. The experimental plots consisted of four 3-m long rows, spaced 0.5 m apart and 15 seeds sown per meter.

The following traits were evaluated: days to first flower (DFF) and days to flowering (DTF) were collected on all seasons and locations. DFF was measured as the number of days from planting until at least one plant presented a flower. DTF was measured as the number of days from planting to when at least 50% of the plants in a plot (replicate) had at least one open flower.

The DNA samples were genotyped using the Vera Code1 BeadXpress (Illumina, San Diego, CA, USA) platform at the Embrapa Biotechnology Laboratory (Goiânia, GO, Brazil). A set of 384 SNP markers, validated by a previously identified Prelim file (https://icom.illumina.com/Custom/UploadOpaPrelim/)

for *Phaseolus vulgaris*, was selected to compose the panel of SNP markers of Oligo Pool Assay (OPA). During the procedure for SNP detection, three oligonucleotides were used for each of the variants of the same SNP and the third specific locus attached to the 3 'region of the DNA fragment containing the target SNP, generating a single allele specific fragment. The genotype call was performed using the Genome Studio software 1.8.4 version (Illumina, San Diego, CA, USA), with Call Rate values ranging from 0.80 to 0.90 and GenTrain ≥ 0.26 for clustering of SNPs. Analyses were performed to group the SNP alleles of each line, based on the signal intensities of the Cy3 and Cy5 fluorophores.

## 2.2. Phenotypic Data Analysis

The results of the analysis of variance to phenological traits for DFF and DTF data of the Brazilian bean cultivars have already been presented by Nascimento et al. [7]. The model adopted was as follows:

$$y_{ijk} = m + g_i + a_j + ga_{ij} + b_{k(j)} + \varepsilon_{ijk},\tag{1}$$

whereby $Y_{ijk}$ is the observed phenotype; m is the general average; $g_i$ is the genotype effect (random; $i = 1, 2, 3, \ldots, 80$), $a_j$ is the effect of environment (fixed; $j = 1$ to 4); $ga_{ij}$ the effect of the interaction of genotype $i$ with environment $j$ (random), $b_{k(j)}$ is the effect of the block (random; $k = 1, 2, 3$), and $\varepsilon_{ijk}$ is the experimental error, Normally and Independently Distributed (NID). After the model fitting for each trait, genetic parameters (heritability and correlations) were estimated for the flowering traits. The variance homogeneity test was performed through Bartlett's test. On the other hand, the normality test was via $X^2$ and Lilliefors [23]. The means were grouped by the Scott and Knott test [24].

## 2.3. Prediction Models for Genomic Estimated Breeding Values

Before fitting the genomic prediction models, the adjusted phenotypes ($Y_i^*$) were obtained as the sum of random effects (genotypes and error). The general genomic model is given by:

$$Y_i^* = \mu + \sum_{m=1}^{p} X_{im}\beta_m + e_i\tag{2}$$

where $Y_i^*$ is the observed phenotypic value of the $i$th individual, which were obtained as the sum of random effects (genotypes and error); $\mu$ is the grand mean; $X_{im}$ is the incidence of the $m$th SNP in the $i$th, p is the total number of SNPs, $\beta_m$ is the estimated random additive marker effect of the $m$th marker $\sim N(0, \sigma_g^2)$, and $\varepsilon_i$ is the residual error term $\varepsilon_i \sim N(0, \sigma_g^2)$ associated with $Y_i^*$. The genomic estimated breeding value was obtained using RR-BLUP [8]. The models were implemented for analysis in Genes software [25] integrated with R using the package RR-BLUP [26].

## 2.4. Artificial Neural Networks

Two ANN models were used to predict the individual genetic merits. Specifically, the Multilayer Perceptron and Radial basis function network approaches were used.

### 2.4.1. Multilayer Perceptron (ANN—MLP)

A feed-forward back propagation multilayer perceptron network was defined considering two hidden layers, activation functions logistic sigmoid or hyperbolic tangent. The number of neurons in each layer varying from one to four and the maximum number of iterations was equal to 5000. The ANN-MLP that presented a lower prediction error was chosen. The matrix of molecular markers was considered as input information, so that the output layer of the ANN-MLP returns the vector of genomic estimated breeding values (GEBV). The backpropagation algorithm was used to train the ANN-MLP. The architecture of the ANN is shown in Figure 1. The model ANN-MLP was implemented for analysis in Genes [25] integrated with Matlab [27].

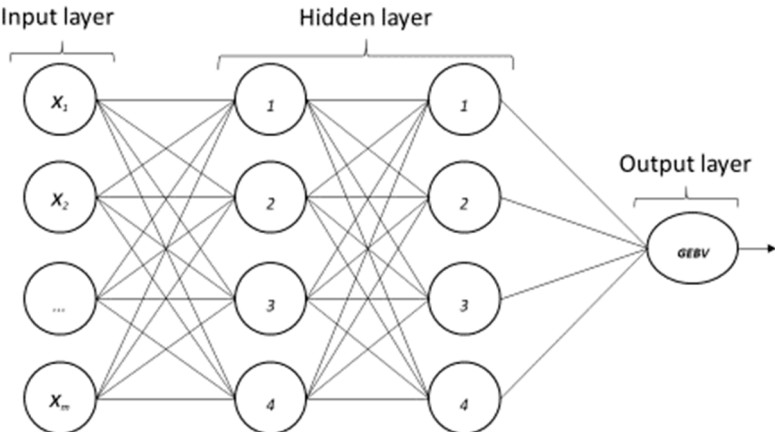

**Figure 1.** Schematic of Artificial Neural NetworksMultilayer Perceptron. Two intermediate layers ($n_{i1}$ and $n_{i2}$) constituted of *i* neurons ($i = 1, \ldots, 4$). The Artificial Neural Network (ANN) returns the vector of genomic estimated breeding values (*GEBV*).

### 2.4.2. Artificial Neural Networks—Radial Basis Function Network (ANN-RBF)

The ANN-RBF is a three layered feed-forward neural network, where the first layer is linear and only distributes the input signal, while the next layer is nonlinear and uses Gaussian functions (Figure 2).

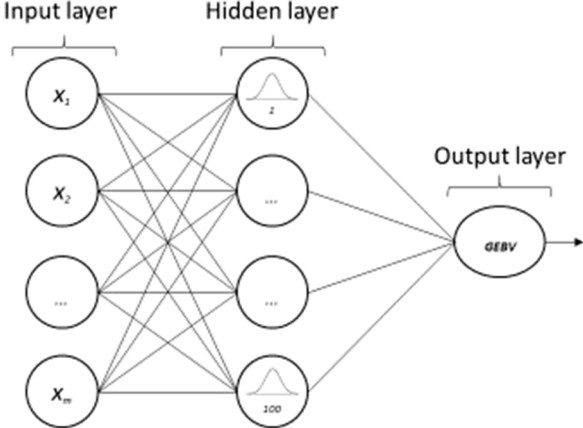

**Figure 2.** Schematic of a Radial Base Function Network. Inputs $X_1$ through $X_{384}$ in the input layer refer to the markers considered in the analyses. A hidden layer considering rays of size r (r ranging from 1 to 80) and consisting of *k* neurons ($k = 1, \ldots, 100$). The RBF returns the vector of genomic estimated breeding values (GEBV).

The ANN-RBF architecture used was feed-forward, with an intermediate hidden layer considering from 1 to 100 neurons with a radius (*r*) ranging from 1 to 80. As for neural network ANN-MLP, the matrix of molecular markers was considered as input information so that the output layer of the RBF returns the vector of GEBV.

### 2.5. Comparison of ANN-RBF, ANN-MLP and RR-BLUP to Estimate GEBV in 5-Fold CV

The mean square error root of the model (MSER), the determination coefficient ($R^2$) and the predictive ability, which is given by the Pearson's correlation between the predicted values and the phenotypes were calculated using a five-fold cross-validation (CV) random process (Figure 3).

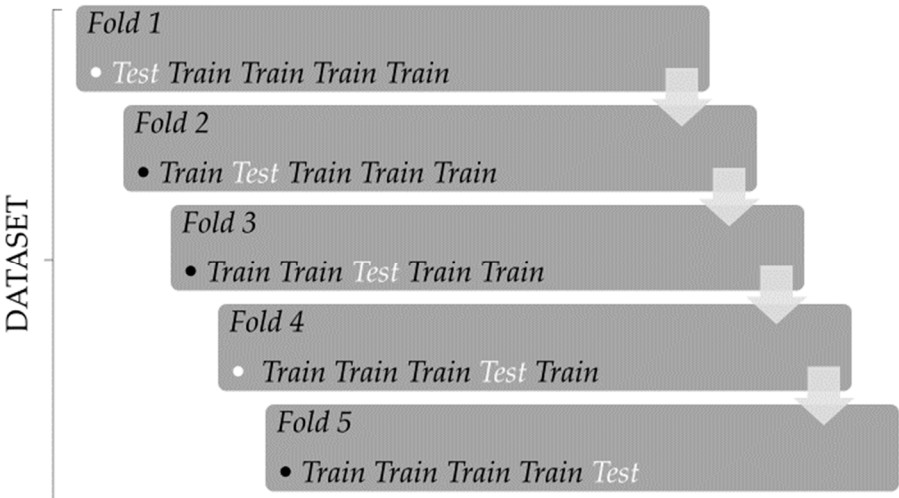

**Figure 3.** Overview of genomic selection with five-fold cross-validation (CV) random process.

RR-BLUP and the ANNs model fittings were carried out using the Genes software [21], which has an integration module with the R software [28] and Matlab [27].

## 3. Results

Data normality and homogeneity were observed, considering $\alpha = 0.05$, by Lilliefors and Bartlett tests, respectively. As observed in [7], the joint analysis of variances showed that there were significant differences between the genotypes, revealing the genetic variability among the cultivars. Estimates of heritability for DTF and DFF were moderate, with $0.58 \pm 0.06$ and $0.49 \pm 0.02$, respectively. Phenotypic and genetic correlation estimates were all positive. Between DFF with DTF, genetic correlation was $0.98 \pm 0.01$, and phenotypic correlation was $0.68 \pm 0.04$.

The predictive ability using computational intelligence-based methodologies, that is, ANN-RBF (DFF: **0.653 ± 0.11** e DFT: **0.961 ± 0.01**) and ANN-MLP (DFF: **0.962 ± 0.001** e DFT: **0.981 ± 0.01**), were superior to those based on RR-BLUP (DFF: **0.561 ± 0.22** e DFT: **0.632 ± 0.13** ) to predict the genetic merit of individuals for flowering traits. The ANN methodologies ANN-RBF (DFF: **0.941 ± 0.001** e DFT: **0.944 ± 0.02** and ANN-MLP (DFF: **0.996 ± 0.001** e DFT: **0.981 ± 0.001**) presented values of $R^2$ higher than the values found by GS (RR-BLUP (DFF: **0.772 ± 0.02** e DFT: **0.841 ± 0.01**)) during the training phase. It is worth mentioning that, for the validation phase, the results obtained by ANN were 90% and 40% times higher than those observed using RR-BLUP for DFF and DFT, respectively (Figure 4—$R^2$). Several authors have used this parameter in order to verify the efficacy of methodologies that involve problems of prediction or classification of simulated populations [29–31] and has also observed efficacy in the use of ANNs. In this case, it is worth noting that ANN-MLP was the methodology that provided predictive abilities above 90%, which quantifies its efficacy (Figure 4).

The genotypes most early flowering (IPR Andorinha, BR-2 Grande Rio, Carioca 1070, IPR Colibri, IAC Imperador, Capixaba Precoce) were the ones that presented the least GEBV (Figure 5—part a).

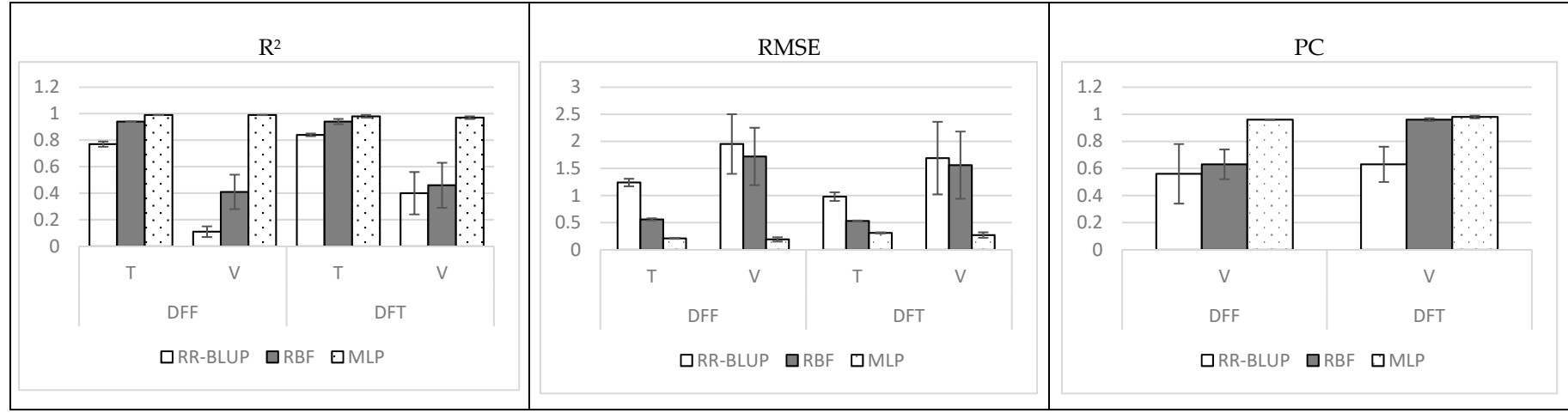

**Figure 4.** Correlation coefficient ($R^2$), root-mean-squared error (RMSE) of training (T) and validation (V) and predictive ability (PC) of the phenological traits for days to first flower (DFF) and days to flowering (DTF) obtained through the GS methodologies: RR-BLUP and ANNs: RBF and MLP for the bean cultivars of carioca and black beans recommended in Brazil between 1960 and 2013.

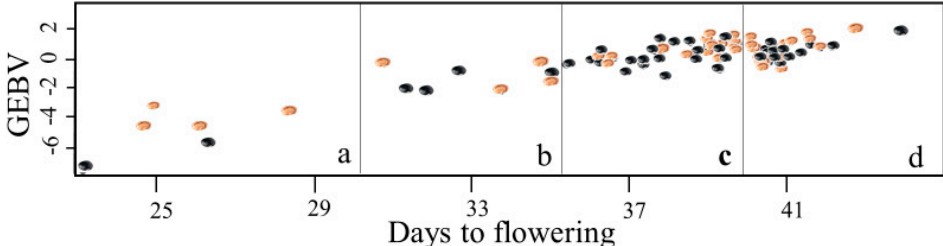

**Figure 5.** Behavior of Carioca (brown beans) and Preto (Black beans) bean cultivars on estimated genomic breeding values (GEBV) and the phenotypic average of phenological traits for days to flowering (DTF). Groups of bean cultivars allocated to the same block (**a**) IPR Andorinha, BR-2 Grande Rio, Carioca 1070, IPR Colibri, IAC Imperador, Capixaba Precoce; (**b**) Moruna, BRS Notável, BRSMG Madrepérola, BRS Majestoso, Diamante Negro, BR 6-Barriga verde, Ouro Negro, BRSMG Talismã; (**c**) Milionário 1732, Onix, BRS Requinte, Varre-Sai, IPR Tuiuiú, BRS Estilo, IAPAR 16, BRSMG Pioneiro, IAC—Carioca Pyatã, VC15, IRAÍ, FT 120, VP 22, IAC Formoso, IAC Tunã, IAC Alvorada, Carioca 1030, Aporé, IPR Eldourado, Xamego, IAC—Carioca Akytá, BRS Esplendor, IAPAR 44, RP1, Rio doce, BR-IPAGRO 1- Macanudo, BRS Expedito, BRS Grafite, BR-3 Ipanema, IAC-Ybaté, BRS Pontal, Carioca 80, IAC-Una, BR 1- Xodó, IAPAR 57, BRS Campeiro, BRS Supremo; and (**d**) BR-Ipagro 2 Pampa, IAC Votuporanga, IPR Gralha, IPR 139, Rico 23, IAPAR 81, IAC-Apuã, Rio Tibagi, IPR Tangará, IPR Saracura, IAPAR 31, BR- IPA 11-Brígida, IPR Uirapurú, SCS Guará, IAPAR 65, IPR Tiziu, IAPAR 20, Iapar 8-Rio Negro, BR- IPA 10, Rudá, Pérola, BRS Cometa, BRS Valente, Rico 1735, IPR Graúna, Preto Uberabinha, IAC Carioca, FT bonito, Campos Gerais for DFT, do not statistically differ by Scott Knott's means clustering test at 5% probability.

## 4. Discussion

The results obtained corroborate the initial expectation that the neural networks, unlike the traditional GS models, allows to capture nonlinear relations from the data information and, in this way, would be able to capture more effectively the non-additive effects associate to the genetic control of flowering traits on a panel of bean cultivars [20,21].

Due to the importance of non-additive effects, several papers have been proposed aiming at semi- and non-parametric models to improve prediction accuracies [32–34]. Overall, GS has been widely studied in and applied to major crop species including both cereals and legumes [35]. However, applications of GS methods using computational intelligence-based methodologies of ANN-RBF and ANN-MLP is still limited. González-Camacho et al. [18], using simulated data, showed that the ANN-RBF model captured epistatic effects. González-Camacho et al. [19] concluded that a Probabilistic Neural Network was more accurate than ANN-MLP for assigning maize and wheat lines. In addition, [14], considering the accuracy of the prediction of leaf rust resistance, showed that methodologies based on Computational Intelligence (including ANN) performs better than G-BLASSO.

It is known that in selfing species, like common bean, non-additive effects, for example epistasis, are expected due to high level of homozygosity [36]. The epistatic interactions have been found considering flowering time traits in barley [37], rice [32], sorghum [38], and cowpea [35]. However, the most used statistical models cannot efficiently characterize or account for epistasis and, therefore, the quantification of the non-additive effects, as epistasis and dominance, has not been fully realized [37,39].

Flowering time is an important adaptive trait in breeding. In this study, our results lead us to believe that that the flowering time variation in 80 common bean cultivars recommended by Brazilian Breeding Programs between 1960 and 2013 (Figure 5) can be due to large and moderate main effects and epistatic loci. Epistatic loci underlie flowering time in both selfing [37,40–42] and outcrossing [43] species.

The importance of ANN in genetic improvement is confirmed in other studies. Coutinho et al. [44], by means of simulated data, compared the prediction methods by ANN and RR-BLUP /GS using correlations between the phenotypic value and genotypic value with the genomic estimated breeding value (GEBV). The results showed superiority of ANN in the prediction of GEBVs in the scenarios with

higher and lower density of markers, parallel to higher levels of linkage disequilibrium and greater heritability. In the characterization of Italian rice cultivars by Marini et al. [45], ANN by Kohonen, used to group data, was able to predict more than 90% of sample sets.

The potential application of ANN as a genetic divergence analysis tool, an important step in the selection of contrasting individuals to be used in breeding programs, is represented in the results found by Barbosa et al. [46]. These authors reported the ANN generating four groups of papaya (*Carica papaya* L.) accesses with 90% of them correctly classified. The ANN was more accurate in predicting corn and soybean yield depending on climatic conditions—coefficient of determination ($R^2$) of 0.77 for corn and 0.81 for soybean—compared to Multiple Linear Regression—$R^2$ of 0.42 for maize and 0.46 for soybeans [47].

Silva et al. [29] applied ANN via simulated traits with 40% and 70% heritability to predict genetic values and gains. The authors identified greater effectiveness in selection using ANN than on the basis of the genotypic mean estimated by maximum likelihood. Higher coincidences between selected and rejected genotypes based on GEBV were also found for ANN than for the genotype mean.

The researcher must choose the methodology that gives them the possibility of quantifying how close the GEBV are to the true value they expect. For regression model adjustments, the literature proposes the use of the root-mean-squared error (RMSE) as the most adequate measure to play such a role [48]. The DFF and DFT traits presented better results, considered $R^2$ and RMSE, than those obtained by RR-BLUP methodology (Figure 4). Methodologies based on neural networks that do not depend on stochastic information tended to be more efficient because these phenotypic traits are obtained by DFF and DFT and depend on traditional methodologies based on normality. The variables DFF and DFT had ANN based methodologies that were significantly better when compared to the RR-BLUP methodology.

In the case of the RMSE evaluated from the RR-BLUP methodology for DFF, values 100 times higher than those obtained by ANN were observed. This fact alone validates our hypothesis that ANNs are efficient at GEBV. Considering the superiority of ANNs-MLP in ANN-MLP in **$R^2$**, RMSE, and PC to GEBV of individuals for phenological traits of flowering compared to RBF methodology (Figure 4), estimation of their marker effects with those of RR-BLUP was deemed more appropriate.

In addition, the use of ANNs in the improvement has already demonstrated the great potential of this methodology in obtained GEBV with simulated studies to classification [30,49]; stability and adaptability [50], and even genomic selection studies [13,17].

The DFT of common bean cultivars were, from 25 to 40 days (Figure 5), similar to that reported by Buratto et al. [5] and Ribeiro et al. [51], who observed DFT from 28 to 43 days. The early flowering genotypes (Figure 5, part a) are the same presented by IAPAR [52], Delfini et al. [53], do Vale et al. [54], Chiorato et al. [55], Ribeiro et al. [48], Burrato et al. [5], Souza Filho [56].

The DFT is the characteristic that has been used by breeders to evaluate precocity in common bean [51,57]. This character presents high heritability, as well as a positive and high magnitude correlation with the physiological maturation of the grains [58]. Cultivars of common beans, under normal conditions and with well-distributed rains, produce less than the normal cycle; however, its use in certain situations has advantages. During the water period, the cultivation of the early stages minimizes the risks of coinciding the flowering with the period of high temperatures and the harvest with the rainy season [59]; according to these authors, in the cultivation of drought, early cultivars can produce more than the normal cycle, when the rains are concentrated more in the initial phase of the crop. However, early flowering cultivars of beans are more suitable for autumn–winter cultivation. The DFT is the characteristic that has been used by breeders to evaluate precocity in common bean [51,57]. The results obtained in this work can be used to selected genotypes and test them in the field. Thus, it will be possible to validate the model in practice.

## 5. Conclusions

The artificial neural network was able to predict genetic merits by means of genomic estimated breeding values (GEBV) of individuals for traits associated with the flowering time (DTF and DFF) in common bean. The ANN's approaches presented higher predictive ability compared with those obtained by RR-BLUP.



**Author Contributions:** Conceptualization, C.D.C. and R.D.S.R.; methodology, C.D.C., L.D.B., J.E.d.S.C., P.C.S.C., M.N. and R.D.S.R.; software, C.D.C.; formal analysis, C.D.C., M.N. and R.D.S.R.; investigation, L.D.B., J.E.d.S.C., P.C.S.C., R.D.S.R. and V.Q.C.; writing—original draft preparation, C.D.C., R.D.S.R. and M.N.; writing—review and editing, C.D.C., L.D.B., J.T.d.S., J.E.d.S.C., P.C.S.C., R.D.S.R., V.Q.C., M.N. All authors have read and agreed to the published version of the manuscript.

**Funding:** This research was funded by CAPES, CNPq, FAPEMIG, and FUNARBE.

**Acknowledgments:** We would like to show our gratitude to Gabi Nunes Silva, Isabela de Castro Sant'Anna and Ithalo Coelho de Sousa for sharing their knowledge during the manuscript conception.

**Conflicts of Interest:** The authors declare no conflict of interest.

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
