# Peer review of "Artificial Neural Networks in the Prediction of Genetic Merit to Flowering Traits in Bean Cultivars"

_agriculture, doi:10.3390/agriculture10120638_

Round 1

Reviewer 1 Report

Thanks for addressing all comments. I think it has been well improved.

Reviewer 2 Report

Thank you for these updates. I am sure this paper will drive interest in the quantitative genetics/plant breeding.

This manuscript is a resubmission of an earlier submission. The following is a list of the peer review reports and author responses from that submission.

Round 1

Reviewer 1 Report

It was an interesting paper to compare Artificial Neural Networks method with RR-BLUP in the GS context. I think most of the paper looked Ok to me, but I would like to see some improvement in some parts.

1) I think RR-BLUP means Rigid for the first "R" instead of "Random". Please correct it across the paper.

2) L106-116: the model terms were confused for me. You mentioned 4 locations, 3 replicates each in the previous part. But in the model term, you used block j, environment j, block k. These should be consistent. You mentioned variance homogeneity test and normality test here, but I didn't see the results for these tests. Also better list a reference for Liliefors which some readers wouldn't know it.

3) 2.5 and 2.6 sessions should be 2.4.1 and 2.4.2? As both were subtitle of 2.4 ANN for me.

4) Figures 1, 2, 3 and 5 looks too small, need to improve figure quality.

5) L164, 80% was training, 20% for validation, is this design for ANNs method only? It seems 80% is a quite large proportion of data the training data set, as we know the bigger the training set is, the better the predictive ability will be.  Does this apply to RR-BLUP as well? Because in Figure3, you also mentioned 5-fold CV, I could understand you calculated predictive ability (PA) by 5-fold CV for RRBLUP, did you also use 5-fold cross-validation for ANNs? or you just used 20% validation set to get PA?  L181-183 also very confused me, you mentioned the square of PA here, but I didn't see any results for this. in the results session. I think you need to spend effect to make all these paragraphs much clearer.

6) L187, individual and joint analysis? Did you mean univariate and bivariate analysis? 

7) L193, 'predictive capacity', I prefer to use 'predictive ability' across the paper. 

8) Figure 5, the Y-axis title is 'EVGB'? Should be 'GEBV'? The dots have two colours, what did it mean for each colour?

5) L168, of an 'access'? not sure what you mean for 'access'

6) for Figure 3, I have some trouble to understand it. 

Reviewer 2 Report

Silva Rosado and colleagues provide novel experimental data that demonstrate the use of ANN in prediction of genetic gains in beans. The authors compare two ANN’s - Multilayer Perceptron (ANN - MLP)  and Artificial Neural Networks - Radial basis function network (ANN-RBF)  to the gold standard RR-BLUP.  Flowering time data was collected as days to first flower (DFF) and days to flowering (DTF) were used in the model.   The use Correlation coefficient (R2), root-mean-squared error (RMSE) of training (T) and validation (V) and predictive capacity (PC) to compare the model fit.  This work can emerge as an asset to wide range of tools used in making breeding decisions.  All parts of the manuscript appear written well.  

Comments:

  1. More information about the ANN in the context of analytics used in the breeding/genomic selection.
  2. Please reference other studies that have explored ANN’s in other species or traits. This could be revisited in the Discussion sections.
  3. Please explain advantages of RBF over MLP
  4. Please include a figure showing the distribution of phenotypic data and variability statistics. Were there any outliers? Do ANNs handle outliers efficiently?
  5. Please include future lines of work on how to validate the model both theoretically and empirically to measure how robust the outcomes of the model.